# Impact Resistance of Ultra-High-Performance Concrete Composite Structures

**DOI:** 10.3390/ma16237456

**Published:** 2023-11-30

**Authors:** Huijun Ning, Huiqi Ren, Wei Wang, Xiaodong Nie

**Affiliations:** 1Institute of Defense Engineering, Academy of Military Sciences (AMS), People’s Liberation Army (PLA), Luoyang 471023, China; srs199916@126.com; 2School of Civil Engineering and Architecture, Henan University of Science and Technology, Luoyang 471023, China; m13170328536@163.com; 3Key Laboratory of Impact and Safety Engineering, Ningbo University, Ministry of Education, Ningbo 315211, China

**Keywords:** UHPC, impact resistance, composite structure, reinforcement ratio

## Abstract

Ultra-high-performance concrete (UHPC) is a cement-based material with excellent impact resistance. Compared with traditional concrete, it possesses ultra-high strength, ultra-high toughness, and ultra-high durability, making it an ideal material for designing structures with impact resistance. The research on the impact resistance performance of UHPC and its composite structures is of great significance for the structural design of protective engineering projects. However, currently, there is still insufficient research on the impact resistance performance of UHPC composite structures. To study the impact resistance performance, experiments were conducted on UHPC targets using high-speed projectiles. The results were compared with impact tests on granite targets. The results indicated that when subjected to projectile impact, the UHPC targets exhibited smaller surface craters compared with the granite targets, while the penetration depth was lower in the granite targets. Afterwards, the process of a projectile impacting the UHPC composite structure was numerically simulated using ANSYS 16.0/LS-DYNA finite element software. The numerical simulation results of penetration depth and crater diameter were in good agreement with the experimental results, which indicates the rationality of the numerical model. Based on this, further analysis was carried out on the influence of impact velocity, impact angle, and reinforcement ratio on the penetration depth of the composite structure. The results show that the larger the incident angle or the smaller the velocity of the projectile is, the easier it is to deflect the projectile. There is a linear relationship between penetration depth and reinforcement ratio; as the reinforcement ratio increases, the penetration depth decreases significantly. This research is of great significance in improving the safety and reliability of key projects and also contributes to the application and development of ultra-high-performance materials in the engineering field.

## 1. Introduction

In recent years, military actions and terrorism have become increasingly frequent worldwide, posing serious threats to buildings and facilities in the field of protection engineering [1]. Therefore, research on high-strength protective materials and structures has always been a hot topic in the field of engineering protection. As a new type of cement-based material, ultra-high-performance concrete (UHPC) has excellent compressive and tensile strength as well as impact resistance, making it the most promising protective material for military and civilian protective structures against impact loads [2,3,4,5,6]. Granite is a widely distributed natural stone material with high compressive strength and excellent hardness and durability, making it effective in resisting bullet penetration. Therefore, using granite in the walls, floors, and other protective structures of buildings can enhance their ability to resist projectile penetration [7].

Extensive experiments and numerical simulation studies have been conducted on the impact resistance properties of ordinary concrete. However, relatively less research has been conducted on the penetration resistance of UHPC and rock [8,9,10,11,12,13]. Park et al. [14] conducted experimental and numerical simulation studies on the penetration resistance of HPFRCC. The research showed that with an increase in projectile velocity, the penetration depth and crater diameter increased. Additionally, as the target strength increased, there was a slight decrease in penetration depth. Kim [15] conducted penetration tests on five concrete targets of varying strengths using conical and hemispherical projectiles. The research indicated that the penetration depth decreased as the concrete strength increased. Compared with normal-strength concrete, the penetration depth normalized by kinetic energy decreased when high-strength concrete and UHPC were used. However, as the concrete strength increased, the descending slope decreased. Forrestal et al. [16,17,18] conducted impact experiments on ordinary concrete targets using ogive-nose projectiles of different diameters and velocities. Based on the experimental results, they proposed a semi-empirical formula for predicting the penetration depth. Wang [19] performed high-speed impact experiments on UR50 ultra-early-strength concrete targets using a 35 mm smooth bore cannon and 35CrMnSiA reduced-ratio ogive-nose projectile. They used the ANSYS 16.0/LD-DYNA finite element software to investigate the effects of compressive strength and projectile impact velocity on the depth of penetration (DOP) of UR50 ultra-early-strength concrete. Based on numerical simulation data and experimental results, they revised and validated the Berezan empirical formula. In Radoslav’s [20] study, the response of several UHPFRC panels to projectile impact was investigated. The experiments showed that implementation of the fibers enhanced the response to the projectile impact; however, more than 2% of the fibers in the mixture had no additional positive effect on either the penetration depth or the crater diameter. Tarek [21] conducted projectile penetration tests on normal concrete and UHPC. The test results indicated that the type and content of fibers have little influence on the penetration depth. However, the fiber contents in the concrete lead to smaller crater volumes and, thus, reduce the spalling and scabbing damage. Lee [22] conducted experimental and numerical studies on the structural response of High-Performance Fiber-Reinforced Cementitious Composite (HPFRCC) panels with a matrix strength of 180 MPa and fiber content ranging from 1.0% to 3.0% under projectile impact. Based on this, a finite element model was established using LS-DYNA software to conduct a parametric study and discussion on the mechanical properties of HPFRCC panels under impact loads. Liu [23] conducted penetration resistance tests on ordinary concrete and UHPC. The study showed that the penetration depth, crater diameter, and volume loss of UHPC targets are significantly lower than those of ordinary concrete, indicating better penetration resistance performance. Shen et al. [24] conducted impact tests on rock using projectiles of different diameters. The results revealed that, compared with concrete, intact rock exhibits better penetration resistance performance. Zhang et al. [25] conducted penetration tests on granite targets using a Φ57 mm compressed gas gun, and the results showed that granite has better resistance capabilities against high-speed penetration compared with concrete. Huang [26] utilized ANSYS 16.0/LS-DYNA finite element software to study the penetration resistance of granite and obtained an empirical formula for the penetration depth of granite.

Although rocks have better penetration resistance, they are more difficult to process than concrete. Rocks are naturally occurring materials and, unlike concrete, they are difficult to shape or mold. Concrete can be poured into molds, making it easy to achieve the desired shape. In comparison, working with rocks typically requires specialized tools and techniques such as drilling, cutting, or blasting to achieve the desired form or size. In this regard, Yang et al. [27] conducted penetration resistance tests on a granite-reinforced concrete composite target. The results showed that the addition of granite significantly improved the penetration resistance of the target. Sonhan et al. [28] proposed a novel multi-layer composite structure and investigated the dynamic characteristics of multi-layer composite materials under projectile impact loads. The study showed that compared with its reinforced concrete monolayer counterpart, the proposed composite material target exhibited enhanced resistance to penetration and reduced damage. Zhai [29] conducted penetration resistance tests and numerical simulations on armored steel/ceramic/UHPC composite structures. The study showed that compared with ordinary concrete targets, the penetration resistance of armored steel/ceramic/UHPC composite targets improved by 77.4%. Liu [30] conducted penetration resistance tests and numerical simulations on ceramic ball/UHPC composite targets. The study showed that compared with traditional UHPC targets, ceramic ball/UHPC composite targets can significantly enhance the penetration resistance performance.

The aforementioned research mainly focuses on the anti-penetration performance of granite, UHPC, and ceramic/armored steel/UHPC composite structures. However, there is relatively less research on the anti-penetration performance of UHPC/granite composite structures. Furthermore, there are limitations to the application of ceramics and armored steel in protective engineering due to their high cost. Based on previous research on the anti-penetration performance of UHPC and granite, this paper compares the anti-penetration performance of UHPC and granite under the same penetration conditions. A UHPC/granite anti-penetration composite structure is designed, and the anti-penetration performance of the composite structure is studied using ANSYS 16.0/LS-DYNA finite element software. The effects of penetration speed, projectile hit angle, and target reinforcement ratio on the penetration depth are systematically investigated. A flow chart for the methodology of this paper in provided in Figure 1.

## 2. Materials and Methods

### 2.1. Material

The mix proportions of the UHPC material in this article is shown in Table 1, mainly composed of ordinary Portland cement (P.O.52.5), quartz sand, quartz powder, silica fume, fly ash, steel fibers, and a superplasticizer. The relevant parameters of steel fibers are shown in Table 2. During the mixing process of UHPC, a dry–wet mixing process is adopted. First, the weighed quartz sand, quartz powder, and steel fibers are poured into the mixer and mixed for more than 5 min. After it is evenly mixed, the weighed cement, fly ash, and silica fume are poured into the mixer and continue to mix for 5 min. Finally, the water and water reducing agent are weighed and poured into the mixer and mixed for about 15 min to form a well-flowing slurry.

Pour the prepared UHPC slurry into the mold; first, pour in half of the height of the slurry, place the mold on a vibrating table, and vibrate for 1 min. Then, pour in the remaining half of the slurry and vibrate for another minute. Finally, smooth the surface of the specimen. Place the specimens in a curing room with a temperature of 20 °C and a humidity of 95% for 28 days of curing. After curing, remove the specimens from the molds and grind the two ends of the loading surface of the specimens to ensure that the flatness of the two end faces of the specimens is less than 5‰.

According to the Chinese Standard GB/T 31387-2015 [31], the UHPC specimens were subjected to quasi-static uniaxial compression tests using the MTS815 testing system. The loading rate was 1.2 MPa/s. The measured standard uniaxial compressive strength of the UHPC was 160 MPa.

### 2.2. UHPC Targets

In the present projectile impact test on UHPC targets, three cylindrical targets of the same size were made, with a diameter of 1300 mm and a thickness of 600 mm, as shown in Figure 2. The manufacturing method and target size were the same as those in reference [32].

### 2.3. Projectile

The projectile material used in the impact test is 35CrMnSiA, with a yield strength of 1300 MPa. The length of the projectile is approximately 307 mm, and its mass is approximately 1000 g. The diameter of the projectile is 30 mm, as shown in Figure 3.

### 2.4. Results and Analysis

The Φ35 mm caliber cannon is used as the experimental launching equipment, and the projectile velocity is controlled between 200 and 350 m/s by adjusting the amount of propellant charge. The test results were compared with the anti-penetration test results of granite mentioned in reference [32], as shown in Table 3.

Figure 4 shows the variation curve of dimensionless depth of penetration (*h*/*d*) with projectile velocity for the experiments in this study and the reference [32]. It can be observed that the penetration depth of both the UHPC and granite targets increases with increasing penetration velocity. At similar penetration velocities, the penetration depth of granite target is significantly smaller than that of UHPC target, indicating that granite performs better in reducing projectile penetration depth.

Figure 5 shows the damage on the surface of UHPC and granite targets [32]. The calculation method for the crater diameter is to take the average of the diameters in four directions, which is shown in Figure 6. It can be observed that the size of the impact crater on the UHPC target is significantly smaller. This is because when the projectile impacts the target, the UHPC material undergoes intense compression and shear deformation. At the same time, the impact generates a shock wave that reflects on the surface of the target, causing the projectile surface of the target to be in a tensile state. UHPC has higher tensile strength compared with granite, which leads to smaller impact craters. In addition, the impact craters in the UHPC target have a more regular circular shape, while the impact craters in the granite target are irregular, and there are also far more cracks in the granite. This is mainly due to the presence of numerous initial pores and cracks in the granite target. When the projectile impacts the target, the target breaks along the weaker planes and separates from the target body, resulting in irregularly shaped impact craters. The shock wave generated by the impact is reflected on the surface of the target plate, causing the surface material of the target plate to be in a tensile state; meanwhile, UHPC has higher tensile strength than granite and, therefore, the crater is relatively small.

Based on the above analysis, it can be concluded that granite performs better than UHPC in reducing projectile penetration depth, while UHPC demonstrates better performance in suppressing impact crater diameter. When UHPC and granite are combined to form a composite structure, it can achieve superior penetration resistance capabilities.

## 3. Numerical Simulation

Based on the analysis in the previous section, in order to further maximize the penetration resistance performance of UHPC and granite, we designed a UHPC/granite composite structure and conducted numerical simulation research. The UHPC/granite composite structure is shown in Figure 7. The outer layer of the composite structure is UHPC, which provides better performance in reducing the crater diameter. The inner layer is granite, which provides better performance in reducing penetration depth. To further enhance the structure’s resistance to penetration, steel reinforcement is embedded within the UHPC layer. In the numerical simulation, the projectile has a length of 1.2 m, a diameter of 117 mm, and a weight of 56 kg.

### 3.1. Material Models

In numerical simulations, the MAT_PLASTIC_KINEMATIC model in LS-DYNA is chosen for the projectile and rebar material models. MAT_PLASTIC_KINEMATIC is commonly used to describe the plastic behavior of metals, and the model can be represented as follows:(1)$$\sigma_{Y}=\left[1+{\left(\frac{\dot{\varepsilon }}{C}\right)}^{1/P}\right]\left(\sigma_{0}+\beta E_{P}\varepsilon_{eff}^{p}\right)$$
where $\sigma_{Y}$ is yield strength; $E_{P}$ is plastic hardening modulus, $E_{P}=E_{t}E/\left(E-E_{t}\right)$; $\varepsilon_{eff}^{p}$ is effective plastic strain; β is hardening parameter; and *C*, *P* are the strain rate parameters.

The rebar parameter values for the MAT_PLASTIC_KINEMATIC model are given in Table 4. The parameters of the MAT_PLASTIC_KINEMATIC model for the projectile material are referenced from [32].

In LS-DYNA, the Riedal–Hiermaier–Thomas [33] (Mat_RHT) model is commonly used for numerical simulations of projectile impact on concrete or rock. The RHT model consists of two parts: the strength model and the state equation. The strength model includes five components: failure surface, elastic limit surface, strain hardening, residual strength surface, and damage. The model comprehensively considers factors such as strain rate effects, pore compaction, strain hardening, and damage in concrete materials, providing a good representation of the response of UHPC and granite under impact loads [34,35]. So, the RHT model is also chosen for the UHPC and granite material models in this paper. The parameters for the RHT model of UHPC are determined using the method described in reference [32], and the parameter values are shown in Table 5. As for the RHT model parameters for granite, they are directly referenced from reference [32].

### 3.2. Numerical Models

After conducting the convergence test, two element sizes for targets and the projectile are selected in the present study. The finite element model of the UHPC/granite composite structure is shown in Figure 8. The element size for the UHPC and granite is set to 2 cm, while the element size for the projectile is set to 1.5 cm. The projectile, UHPC, and granite models are based on eight-node solid elements (SOLID164). The steel reinforcement is modeled using beam elements (BEAM161). The contact between the steel reinforcement and UHPC is defined using the *CONSTRAINED_LAGRANGE_IN_SOLID keyword. In the simulation, automatic face-to-face contact is used between the projectile and UHPC/granite. This type of contact is implemented by setting the *CONTACT_EROSION_SURFACE_TO_SURFACE keyword in the finite element analysis. When the projectile comes into contact with the surfaces of UHPC and granite, the program automatically detects and considers the contact forces and behaviors between them.

### 3.3. Validation of the Numerical Models

To verify the accuracy of the numerical model, we manufactured a UHPC/granite composite structural target plate, according to the configuration shown in Figure 8, and conducted projectile penetration tests. The comparison between the test results and the numerical simulation results is shown in Table 6. It can be observed that the numerical simulation results are very close to the test results, with an error of approximately 6.9%.

The damage factor *D* is commonly used in numerical simulations under the RHT model to reflect the damage caused by projectiles when penetrating granite targets; $0\leq D=\sum \left(\Delta \varepsilon_{p}/\varepsilon_{p}^{fail}\right)\leq 1$. When *D* = 0, the material is considered intact; when *D* = 1, the material is considered completely damaged. Figure 9 shows the surface damage of the UHPC/granite composite structure under projectile impact, and it can be observed that there is good agreement between the numerical simulation results and the experimental data.

## 4. Results and Discussion

### 4.1. Effect of Impact Angle

The angle of penetration has a significant impact on the depth of penetration. In general, the same projectile will produce different penetration depths at different angles of penetration. When the projectile penetrates at a perpendicular angle to the target surface, the penetration depth is maximum. As the angle of penetration becomes more oblique, the penetration depth decreases. This is due to the resistance of the target material that the projectile encounters during penetration. A larger angle of penetration requires more energy to overcome the resistance of the target material, resulting in less energy available for penetration, thus reducing the penetration depth.

In order to analyze the effects of projectile impact angle on penetration depth and trajectory deviation, the penetration process of a projectile impacting a composite structure at angles of 0°, 10°, 20°, and 30° are simulated, as shown in the Figure 10. The simulated results are shown in Table 7.

According to the numerical simulation results in Table 7, we produced Figure 11, which shows the variation in penetration depth with velocity for projectiles penetrating a composite target at different impact angles. It can be observed that as the impact angle increases, the penetration depth gradually decreases. When the projectile penetrates at a perpendicular angle to the target surface, the penetration depth is maximum. For oblique penetration at different impact angles, the penetration depth approximately follows a linear relationship with penetration velocity. We define the ratio of penetration depth at different impact angles to penetration depth at an impact angle of 0° as the equivalent coefficient *e_a_* of the impact angle; the curve of *e_a_* changing with the impact angle is shown in Figure 12. The curve of the impact angle equivalent coefficient *e_a_* with respect to the impact angle is shown in Figure 12. It can be observed that the impact angle equivalent coefficient increases gradually with the increase in the impact angle. The difference in the equivalent coefficients for different penetration velocities at the same impact angle is not significant. A linear correlation is found between the impact angle equivalent coefficient and the impact angle, as shown in Equation (2).
(2)$$e_{a}=1-0.0105\times \theta$$

Figure 13 shows the trajectory curves of projectiles penetrating a composite structure at a 30° impact angle under different velocities. In this case, *h*_x_ represents the horizontal displacement of the projectile after penetrating into the composite structure, while *h*_y_ represents the vertical displacement of the projectile within the composite structure. From the figure, it can be seen that the penetration trajectories deviate to varying degrees at different penetration velocities. In the initial phase of penetration, the projectile’s motion paths almost overlap, and the deflection becomes greater as the velocity decreases.

Figure 14 illustrates the variation in penetration trajectories at different impact angles when the penetration velocity is 500 m/s. It can be observed from Figure 14 that in the initial stage of projectile impact, the absolute value of the slope of the curve is relatively large, indicating that the lateral displacement of the projectile is small and the deflection is not significant. As the penetration process progresses, the absolute value of the slope of the trajectory curve gradually decreases, indicating that the lateral displacement of the projectile increases. The trajectory undergoes significant deflection. In the final stage of penetration, the slope of the trajectory curve remains relatively constant, indicating that the projectile penetrates the target in a relatively stable posture until the penetration velocity approaches zero. According to the motion trajectories of the oblique-penetrating projectile head shown in Figure 13 and Figure 14, the penetration process can be divided into three stages. In the initial stage of penetration, as the projectile head penetrates into the target, the contact cross-sectional area between the projectile and the target increases. This leads to a significant increase in resistance force acting on the projectile, causing gradual damage to the surface of the target. In the intermediate stage of penetration, when the projectile collides with the target, it causes damage and tensile failure to the UHPC medium near the contact area of the projectile. This results in a decrease in resistance between the projectile and the target. Additionally, due to the asymmetry in the contact area between the projectile head and the target, the projectile experiences an asymmetric force, leading to a certain degree of deflection. In the final stage, when the velocity of the projectile decreases to a certain level and the collision pressure is insufficient to damage the concrete, the projectile no longer has sufficient kinetic energy to penetrate further. As a result, the projectile eventually come to rest within the concrete medium.

### 4.2. Effect of Reinforcement Ratio

Adding steel reinforcement to the UHPC/granite composite structure can further enhance its resistance to penetration. On one hand, the steel reinforcement provides effective confinement to the UHPC/granite composite target, including toughening and crack arrest capabilities, strengthening the target’s resistance to penetration and reducing the formation of penetrating cracks caused by projectile impact. On the other hand, steel reinforcement itself possesses high strength. During the process of penetrating the UHPC/granite composite target, the deformation and failure of the steel reinforcement consume a significant amount of energy from the projectile when the projectile collides with the steel reinforcement, thereby reducing its destructive effect on the target.

In order to analyze the effect of reinforcement ratio on the penetration resistance of composite structures, a set of calculation models for UHPC/granite composite targets with reinforcement ratios of 4.6%, 3.45%, 2.3%, 1.15%, and 0% were established. The effects of different target reinforcement ratios on the penetration depth were studied at four impact velocities of 200 m/s, 300 m/s, 400 m/s, and 500 m/s. The simulation results are shown in Table 8.

The relationship between penetration depth and impact velocity for UHPC/granite composite structures with different reinforcement ratios is shown in Figure 15. It can be observed that as the reinforcement ratio increases, the penetration depth significantly decreases. The dimensionless penetration depth of composite structures with different reinforcement ratios shows a linear relationship with velocity. In order to facilitate the comparison of the penetration resistance between reinforced concrete targets and plain concrete targets, the concept of an equivalent coefficient was introduced in reference [36]. This coefficient represents the ratio of the penetration depth of a reinforced concrete target to the penetration depth of a plain concrete target under the same conditions of velocity and projectile geometry. Figure 16 presents the variation curve of the equivalence coefficient *e_r_* with reinforcement ratio under different penetration velocities. It can be observed that the value of the equivalence coefficient *e_r_* decreases as the reinforcement ratio increases at a given penetration velocity, indicating better penetration resistance performance of the structure. The reinforcement ratio and the equivalence coefficient *e_r_* exhibit a linear relationship.

The variation curve of the equivalence coefficient *e_r_* with penetration velocity is shown in Figure 17. It can be observed that as the projectile impact velocity increases, the equivalence coefficient gradually decreases. This indicates that at higher penetration velocities, the reinforced composite structure has a more significant energy consumption effect on the projectile compared with the plain concrete composite structure. When dealing with high-speed penetration projectiles, the penetration resistance capability of the reinforced composite structure is significantly enhanced compared with that of the plain concrete structure.

By performing linear fitting on the equivalence coefficient *e_r_* and penetration velocity, the following equivalence coefficient calculation formulas for four types of reinforced composite structures and the plain concrete composite structure can be obtained:(3)$$e_{r1.15}=1.04-3.97\times 10^{-4}V$$
(4)$$e_{r2.3}=0.983-4.63\times 10^{-4}V$$
(5)$$e_{r3.45}=0.897-3.86\times 10^{-4}V$$
(6)$$e_{r4.6}=0.824-4.4\times 10^{-4}V$$

## 5. Conclusions

This study conducted impact resistance tests on UHPC targets and compared the differences in penetration depth and crater diameter between UHPC targets and granite targets. In order to further enhance the impact resistance performance, a UHPC/granite composite structure was designed. We carried out numerical simulations using ANSYS 16.0/LS-DYNA finite element software to analyze the impact resistance performance of the UHPC/granite composite structure and investigate the influence of impact speed, impact angle, and reinforcement ratio on its impact resistance performance. The main conclusions are as follows:The impact tests were conducted on UHPC targets with velocities ranging from 216 to 340 m/s. Compared with granite targets, UHPC targets perform better in reducing crater diameter, while granite targets have a greater advantage in reducing penetration depth.The numerical models can effectively predict the penetration depth of projectiles into UHPC/granite composite structures. Meanwhile, compared with experimental results, the damage on the target surface obtained by the numerical model is consistent with the experimental observations.For the UHPC/granite composite structure, the penetration depth significantly decreases with the increase in the projectile incidence angle, and there is a linear relationship between penetration depth and impact angle.The greater the impact angle of the projectile, the more likely the trajectory of the projectile is to deflect during the penetration process. At the same time, the higher the velocity of the projectile, the less likely the trajectory is to deflect.For the UHPC/granite composite structure, there is a linear relationship between penetration depth and reinforcement ratio. Furthermore, the equivalent coefficient *e_r_* of the reinforcement ratio is not only correlated with the reinforcement ratio but also related to the penetration velocity.

## Figures and Tables

**Figure 1 materials-16-07456-f001:**
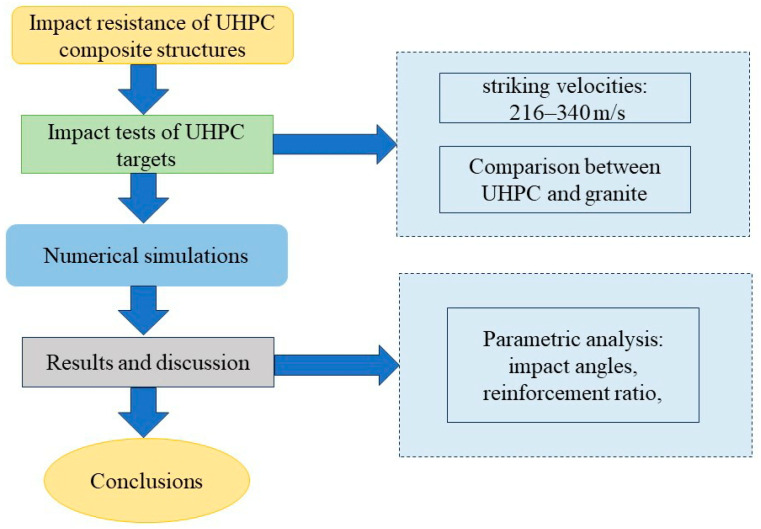
Flow chart for methodology.

**Figure 2 materials-16-07456-f002:**
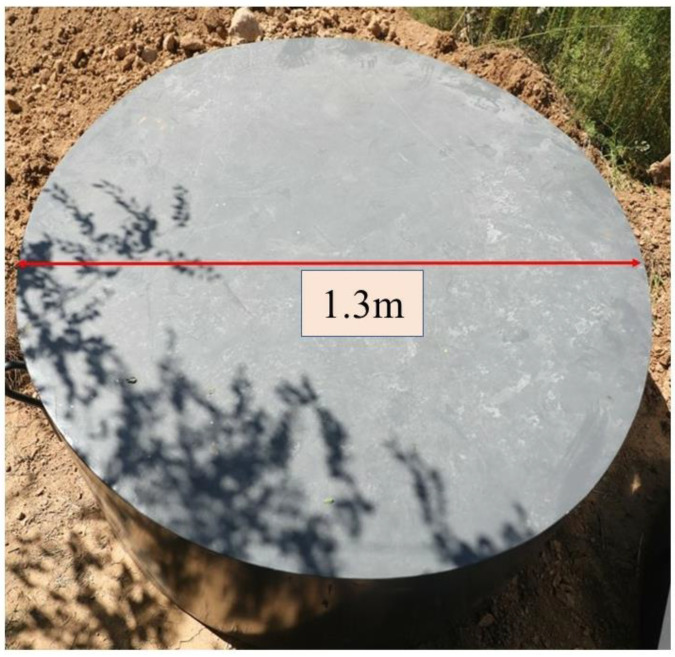
UHPC target.

**Figure 3 materials-16-07456-f003:**
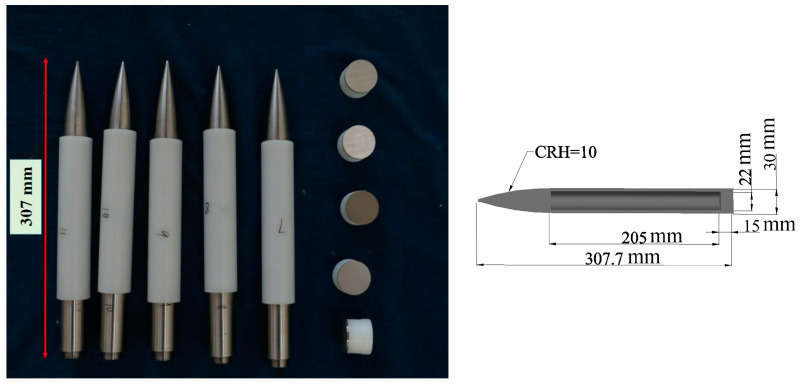
Projectile.

**Figure 4 materials-16-07456-f004:**
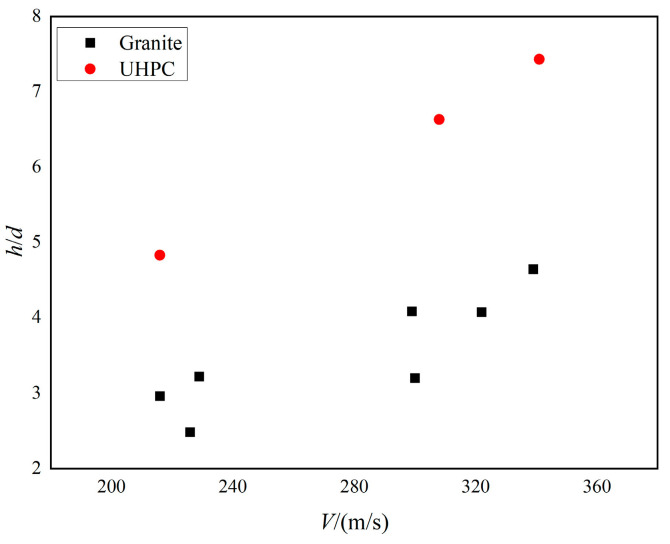
Dimensionless DOP versus striking velocities of projectile. Red: UHPC, Black: Granite [32].

**Figure 5 materials-16-07456-f005:**
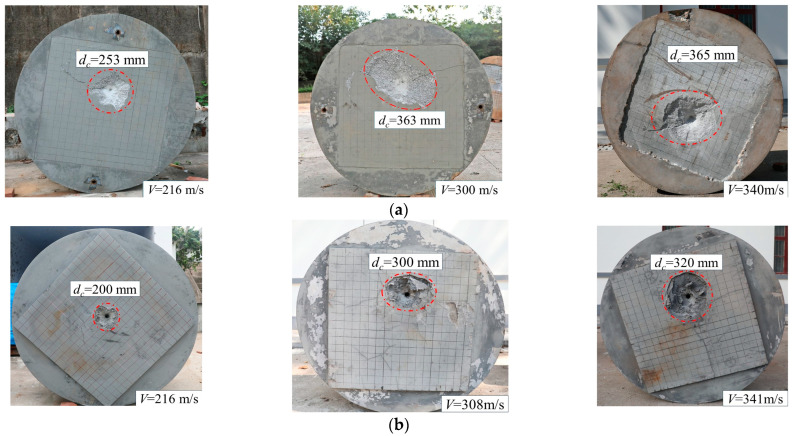
Localized damage of (**a**) granite targets [32] and (**b**) UHPC targets.

**Figure 6 materials-16-07456-f006:**
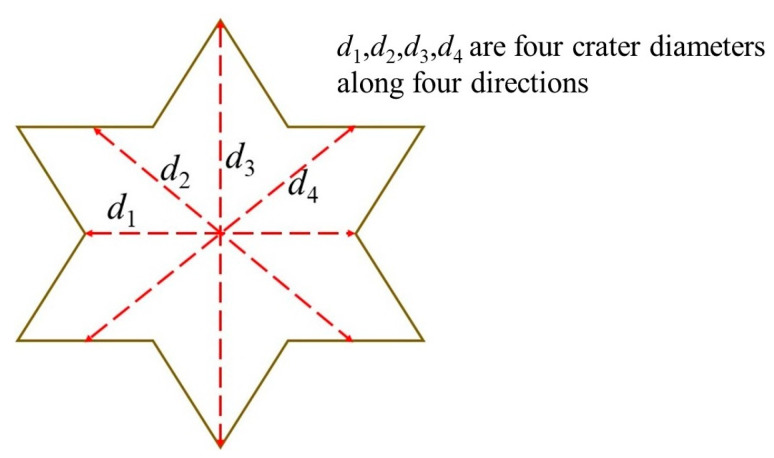
Measurement of average crater diameter.

**Figure 7 materials-16-07456-f007:**
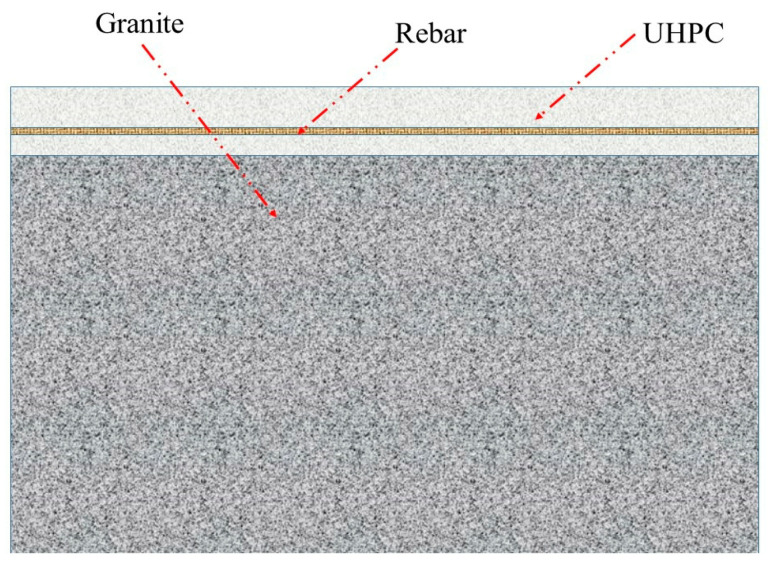
UHPC/granite composite structure.

**Figure 8 materials-16-07456-f008:**
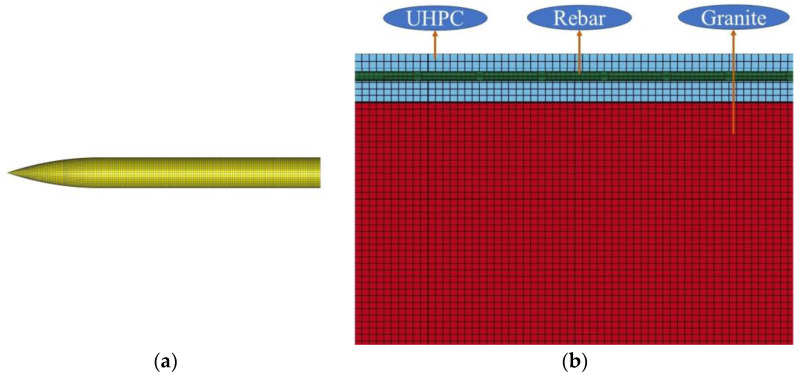
Numerical models: (**a**) projectile; (**b**) targets. Blue: UHPC, Red: Granite, green: Rebar.

**Figure 9 materials-16-07456-f009:**
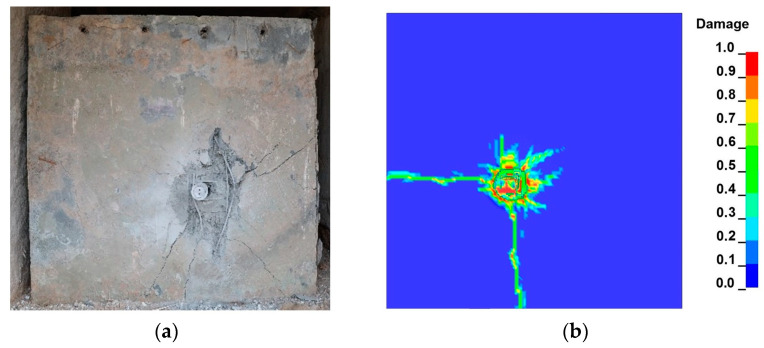
Crater diameter of target: (**a**) experimental; (**b**) simulation.

**Figure 10 materials-16-07456-f010:**
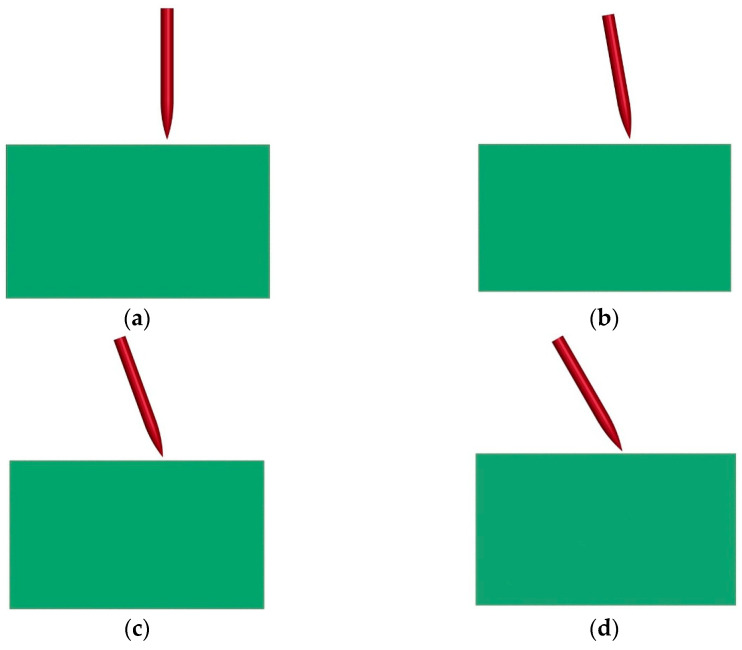
Numerical model of projectile penetrating composite structure at different impact angles: (**a**) 0°; (**b**) 10°; (**c**) 20°; (**d**) 30°. Red: projectile, green: composite structure.

**Figure 11 materials-16-07456-f011:**
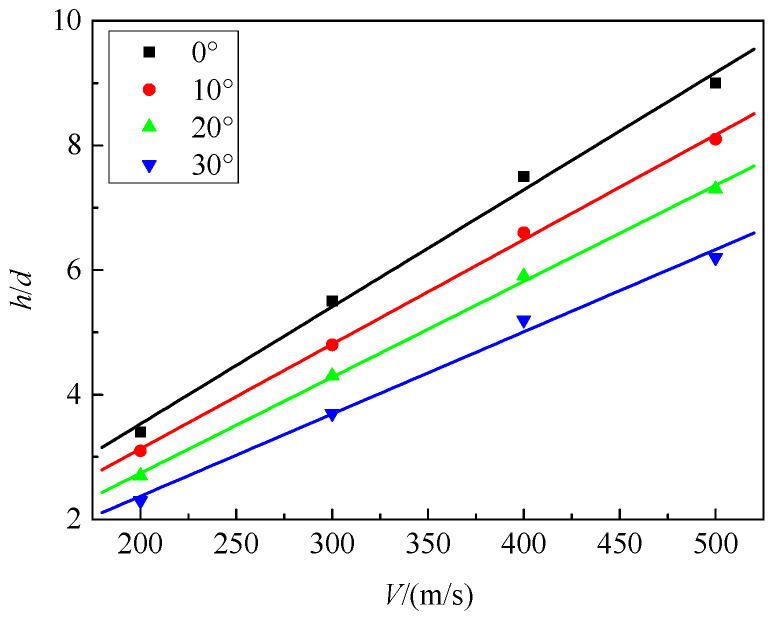
Variation in penetration depth with velocity at different impact angles.

**Figure 12 materials-16-07456-f012:**
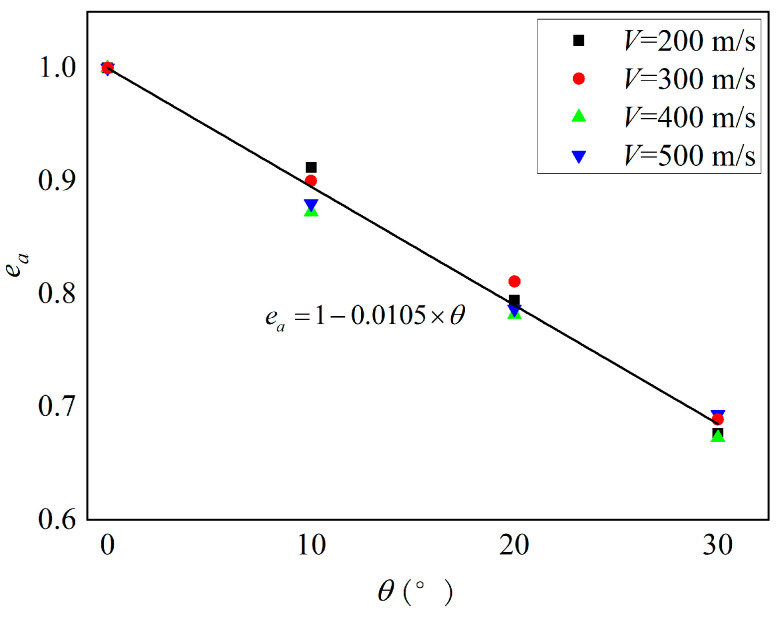
The relationship of impact angle equivalent coefficient with impact angle.

**Figure 13 materials-16-07456-f013:**
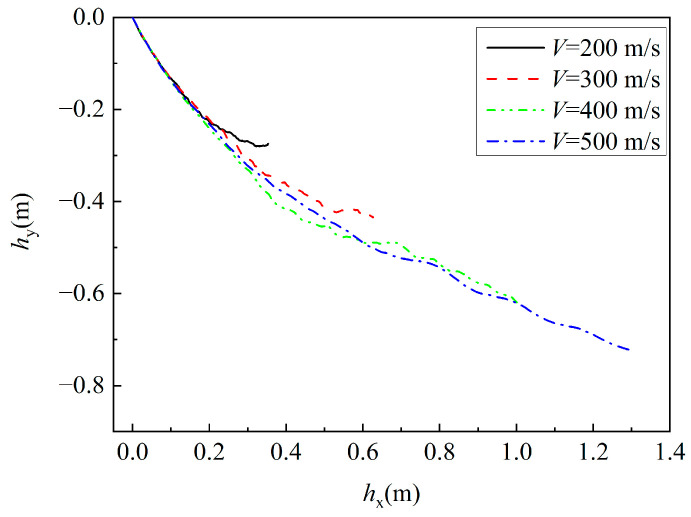
The penetration trajectories of the projectile under different impact velocities.

**Figure 14 materials-16-07456-f014:**
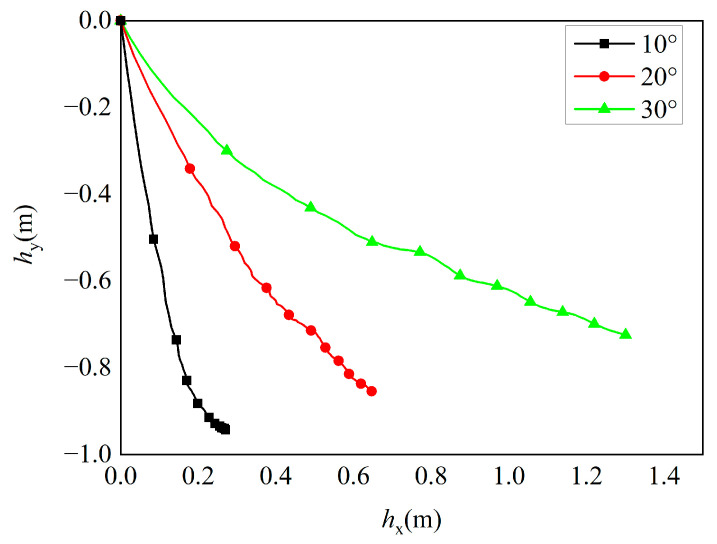
The penetration trajectory at different impact angles.

**Figure 15 materials-16-07456-f015:**
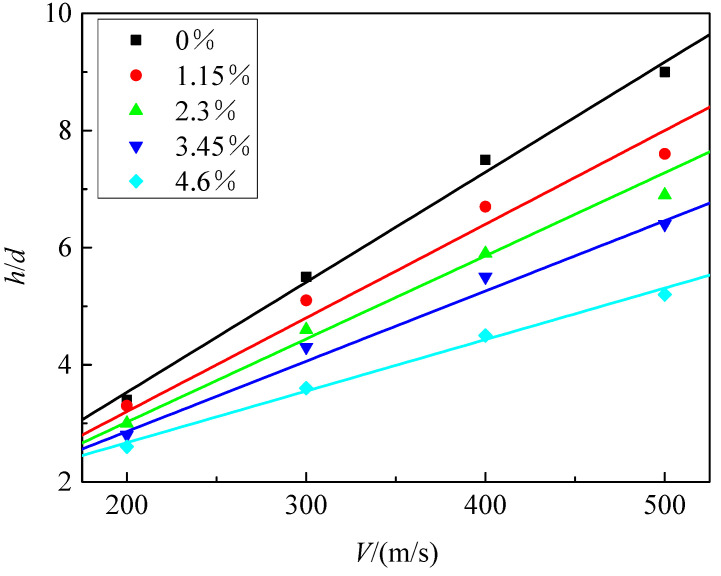
The variation in penetration depth with impact velocity for different reinforcement ratios.

**Figure 16 materials-16-07456-f016:**
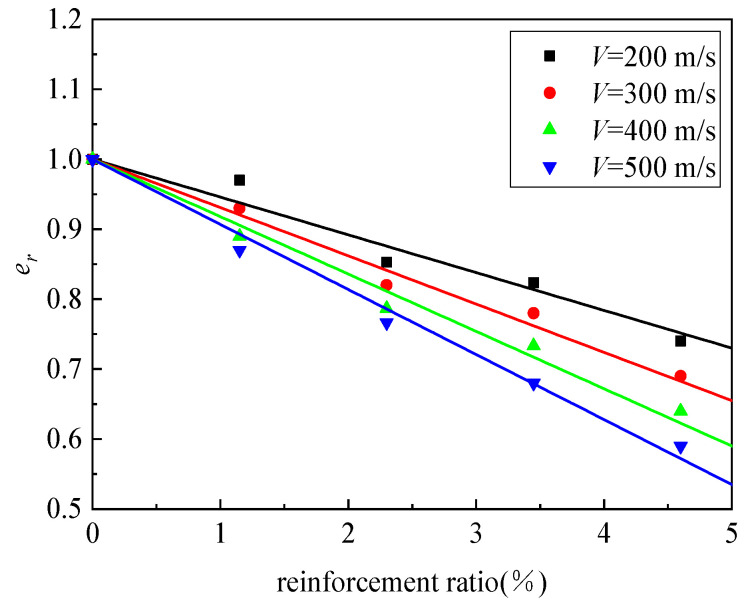
The variation in *e_r_* with reinforcement ratio under different penetration velocities.

**Figure 17 materials-16-07456-f017:**
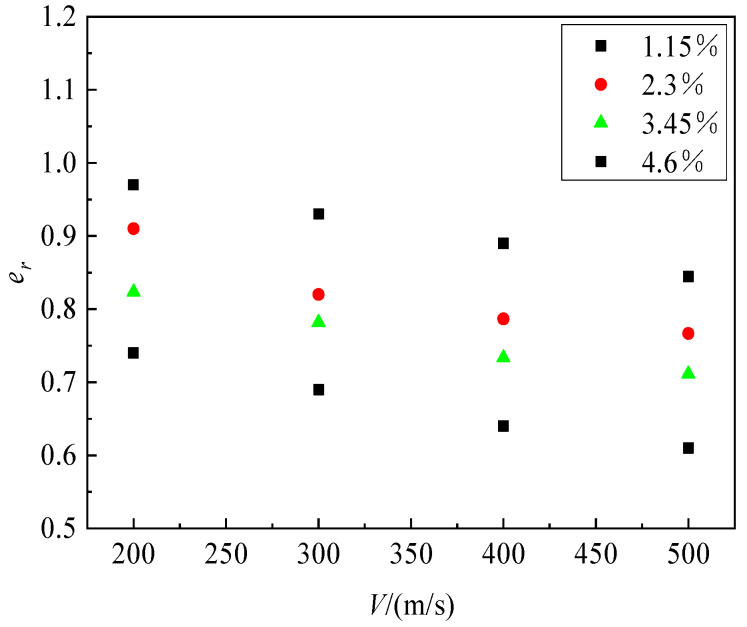
The variation in *e_r_* with impact velocity at different reinforcement ratios.

**Table 1 materials-16-07456-t001:** Mix proportions of UHPC (unit: kg).

Cement	Quartz Sand	Quartz Powder	Silica Fume	Fly Ash	Steel Fibers	Water	Superplasticizer
666	1065	160	160	80	157	135	6.7

**Table 2 materials-16-07456-t002:** Material properties of fibers.

Diameter (mm)	Length (mm)	Strength (MPa)	Young’s Modulus (GPa)	Density (kg/m^3^)
0.2	13	2000	200	7800

**Table 3 materials-16-07456-t003:** Penetration test data.

Types of Target	*d* ^1^ (mm)	*m* ^2^ (kg)	*V* ^3^ (m/s)	Depth of Penetration		Crater Diameter	
				*h* ^4^ (mm)	*h*/*d*	*d_c_* (mm)	*d_c_*/*d*
UHPC	30	1.001	216	145	4.83	200	6.67
	30	1.003	308	199	6.63	300	10
	30	1.003	341	223	7.43	320	10.67
Granite [32]	30	0.999	216	89	2.97	253	8.43
	30	1.002	226	74	2.47	370	12.33
	30	1.003	229	97	3.23	305	10.17
	30	0.999	300	122	4.07	363	12.1
	30	1.004	300	96	3.2	365	12.17
	30	1.005	322	122	4.07	513	17.1
	30	1.003	340	139	4.6	363	12.1

^1^ projectile diameter; ^2^ projectile mass; ^3^ striking velocities; ^4^ depth of penetration.

**Table 4 materials-16-07456-t004:** Model parameters for rebar.

ρ (kg·m^−3^)	*E* (GPa)	$\sigma_{Y}$ (MPa)	ν	$E_{t}$ (MPa)	β	*C*	*P*	$F_{s}$
7800	210	280	0.3	600	1	40	5	0.2

ρ is the density; *E* is the elastic modulus; $\sigma_{Y}$ is the yield strength; ν is Poisson’s ratio; $E_{t}$ is the tangent modulus; β is the hardening parameters; *C*, *P* is the strain rate parameter, and $F_{s}$ is the failure strain.

**Table 5 materials-16-07456-t005:** RHT model parameters for UHPC.

ρ (kg·m^−3^)	*G* (GPa)	$f_{c}$ (MPa)	$B_{1}$	$B_{2}$	$T_{1}$ (GPa)	$T_{2}$	*A*	${\dot{\varepsilon }}^{c}$ (s^−1^)	${\dot{\varepsilon }}^{t}$ (s^−1^)	${\dot{\varepsilon }}_{0}^{c}$ (s^−1^)	${\dot{\varepsilon }}_{0}^{t}$ (s^−1^)
2450	18.5	160	1.22	1.22	44	0	1.6	3.0 × 10^25^	3.0 × 10^25^	3.0 × 10^−5^	3.0 × 10^−6^
$p_{el}$ (MPa)	$\overset{*}{g_{c}}$	$\overset{*}{g_{t}}$	ξ	$D_{1}$	$\varepsilon_{p}^{m}$	$A_{f}$	$n_{f}$	$A_{1}$ (GPa)	$A_{2}$ (GPa)	$A_{3}$ (GPa)	$\beta_{c}$
53.3	0.53	0.7	0.67	0.04	0.008	1.75	0.52	44	49.38	11.28	0.0125
$\beta_{t}$	*B*	*N*	$D_{2}$	$Q_{0}$	*n*	$f_{t}^{*}$	$f_{s}^{*}$	$p_{\mathrm{com}}$ (GPa)	$\alpha_{0}$		
0.0143	0.0105	4.0	1	0.681	0.61	0.0613	0.267	6	1.18		

ρ the density; *G* is the shear modulus; $f_{c}$ is the compressive strength; $B_{1}$, $B_{2}$, $T_{1}$, $T_{2}$ are the parameters for polynomial EOS; $A_{1}$, $A_{2}$, $A_{3}$ are the hugoniot polynomial coefficients; $f_{t}^{*}$ is the relative tensile strength; $f_{s}^{*}$ is the relative shear strength; *A*, *N* are the failure surface parameters; $Q_{0}$, *B* are the lode angle dependence factors; *N* is the porosity exponent; ${\dot{\varepsilon }}_{0}^{c}$ is the reference compressive strain rate; ${\dot{\varepsilon }}_{0}^{t}$ is the reference tensile strain rate; ${\dot{\varepsilon }}^{c}$ is the break compressive strain rate; ${\dot{\varepsilon }}^{t}$ is the break tensile strain rate; $\beta_{c}$ is the compressive strain rate dependence exponent; $\beta_{t}$ is the tensile strain rate dependence exponent; $g_{c}^{*}$ is the compressive yield surface parameter; $\varepsilon_{p}^{m}$ is the minimum damaged residual strain; ξ is the shear modulus reduction factor; $D_{1}$, $D_{2}$ are the damage parameters; $A_{f}$, $n_{f}$ are the residual surface parameters; *A*, *n* are the failure surface parameters; $\alpha_{0}$ is the initial porosity; $p_{el}$ is the crush pressure; and $p_{\mathrm{com}}$ is the compaction pressure.

**Table 6 materials-16-07456-t006:** Comparison of experimental and simulation penetration depth.

Test No	Diameter of the Projectile (mm)	Velocities *V* (m/s)	Reinforcement Ratio (%)	Experiments (m)	Simulation (m)	Error
1	117	300	2.3	0.58	0.54	6.9%

**Table 7 materials-16-07456-t007:** Penetration depth at different impact angles.

θ ^1^ (°)	*d* ^2^ (mm)	*V* ^3^ (m/s)	Depth of Penetration	
			*h* ^4^ (m)	*h*/*d*
0	117	200	0.4	3.4
0		300	0.64	5.5
0		400	0.88	7.5
0		500	1.05	9.0
10		200	0.36	3.1
10		300	0.55	4.7
10		400	0.73	6.2
10		500	0.95	8.1
20		200	0.32	2.7
20		300	0.46	3.9
20		400	0.64	5.5
20		500	0.85	7.3
30		200	0.27	2.3
30		300	0.41	3.5
30		400	0.62	5.3
30		500	0.72	6.2

^1^ impact angles; ^2^ projectile diameter; ^3^ striking velocities; ^4^ depth of penetration.

**Table 8 materials-16-07456-t008:** Penetration depth at different reinforcement ratios.

Reinforcement Ratio (%)	*d* (mm)	*V* (m/s)	Depth of Penetration	
			*h* (m)	*h*/*d*
0	117	200	0.4	3.4
0		300	0.64	5.5
0		400	0.88	7.5
0		500	1.05	9.0
1.15		200	0.35	3.0
1.15		300	0.56	4.8
1.15		400	0.75	6.4
1.15		500	0.90	7.6
2.3		200	0.34	2.9
2.3		300	0.54	4.6
2.3		400	0.69	5.9
2.3		500	0.81	6.9
3.45		200	0.33	2.8
3.45		300	0.50	4.3
3.45		400	0.64	5.5
3.45		500	0.75	6.4
4.6		200	0.30	2.6
4.6		300	0.42	3.6
4.6		400	0.53	4.5
4.6		500	0.61	5.2

## Data Availability

The data used to support the findings of this study are available from the corresponding author upon request.

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
