# Peer review of "Impact Resistance of Ultra-High-Performance Concrete Composite Structures"

_materials, 2023, doi:10.3390/ma16237456_

Round 1

Reviewer 1 Report

Comments and Suggestions for Authors

Impact resistance of ultra-high performance concrete composite structures

General comments

The paper presents the results of an experimental programme on the impact resistance of UHPC, and a preliminary numerical study on a proposed setup composed by granite and UHPC aimed at enhancing the performance under impact.

The paper is generally well written and the flow convincing. This reviewer has some concerns about the novelty of the experimental tests, since impact tests on UHPC specimens are quite spread in the literature, as reported by the authors in the introduction. This should be elaborated on.

I also wonder why no validation of the numerical models has been performed before showing the results on the proposed setup. A validation against the results reported in this manuscript would enhance the quality of the paper. Only after proper validation can a novel composite setup be proposed.

For this reason, the manuscript needs to be re-evaluated after the authors have addressed these concerns and fixed the points reported in the following.

Specific comments

1.      The authors not only compare their results with those on granite reported in [21] (which is fine) but a bit too often the images are similar to those shown there without proper reference in the caption.

2.      Amend caption in figure 2, this is not a projectile.

3.      Line 158, report formula for dimensionless depth of penetration.

4.      Figure 7, “Granite”, not “Grainet”.

5.      Material model acronyms are those of the specific software used, but they should explained more in detail. In particular, the characteristics of the contact are expected to be particularly important.

6.      Line 209: RHT is not defined.

7.      Table 6: specify h and d in the text or in the caption.

Comments on the Quality of English Language

English is good, only some minor amendments are required following a thoroughly reading.

Reviewer 2 Report

Comments and Suggestions for Authors

1. Please refine the abstract to enhance its precision.

2. The article is commendable; however, the introduction would benefit from a broader range of references. Aim to include at least 35 references.

3. The citation for Figure 6 appears to be missing. Kindly review and address this omission.

4. The conclusion section is currently too brief. Please expand it to include approximately five to six succinct points.

5. The manuscript requires editing for English language accuracy. Please attend to this.

Comments on the Quality of English Language

high level of editing required 

Reviewer 3 Report

Comments and Suggestions for Authors

The paper illustrates the results of impact resistance performance tests performed on ultra-high performance concrete (UHPC) targets using high-speed projectiles. The results were compared with impact tests on granite targets.

The second part of the article analyzes the results of a simulation conducted on a finite element numerical model of a UHPC/granite composite structure.

The relationship between the two parts of the research is not adequately illustrated.

Specifically, the size and type of the target elements tested in the laboratory, described in Chapter 2, appear significantly different from the objects of the numerical simulation. The size, mass, and velocity of the projectiles also do not appear comparable.

Therefore, it needs to be clarified whether the numerical model was calibrated to the test results.

The amount, diameter, and arrangement of steel reinforcement within the UHPC layer probably play a key role in damage containment. It is therefore appropriate to provide an in-depth discussion of these elements.

At line 169, it is unclear whether new measurements were conducted on the target elements described in the reference [21] or whether the results of that publication were used directly. The caption in Figure 5 should be adjusted according to this.

Round 2

Reviewer 3 Report

Comments and Suggestions for Authors

The scientific soundness of the test could have been greater by reporting the results of more comparisons of experimental and simulation-derived penetration depths in Table 6. Even in this form, the article is ready for publication.